# B-Cell Regeneration Profile and Minimal Residual Disease Status in Bone Marrow of Treated Multiple Myeloma Patients

**DOI:** 10.3390/cancers13071704

**Published:** 2021-04-03

**Authors:** Robéria Mendonça de Pontes, Juan Flores-Montero, Luzalba Sanoja-Flores, Noemi Puig, Roberto J. Pessoa de Magalhães, Alba Corral-Mateos, Anna Beatriz Salgado, Omar García-Sánchez, José Pérez-Morán, Maria-Victoria Mateos, Leire Burgos, Bruno Paiva, Jeroen te Marvelde, Vincent H. J. van der Velden, Carlos Aguilar, Abelardo Bárez, Aranzazú García-Mateo, Jorge Labrador, Pilar Leoz, Carmen Aguilera-Sanz, Brian Durie, Jacques J. M. van Dongen, Angelo Maiolino, Elaine Sobral da Costa, Alberto Orfao

**Affiliations:** 1Internal Medicine Postgraduate Program, Faculty of Medicine, Federal University of Rio de Janeiro (UFRJ), Rio de Janeiro 21941-617, Brazil; roberiafar@gmail.com (R.M.d.P.); beatrizsalgadobio@gmail.com (A.B.S.); angelomaiolino@gmail.com (A.M.); elainesc.ufrj@gmail.com (E.S.d.C.); 2Cytometry Service, Institute of Paediatrics and Puericultura Martagão Gesteira (IPPMG), Faculty of Medicine, Federal University of Rio de Janeiro (UFRJ), Rio de Janeiro 21941-912, Brazil; 3Translational and Clinical Research Program, Cancer Research Center (IBMCC, USAL-CSIC), Cytometry Service (NUCLEUS) and Department of Medicine, University of Salamanca, Institute of Biomedical Research of Salamanca (IBSAL), 37007 Salamanca, Spain; jflores@usal.es (J.F.-M.); albacorralmateos@usal.es (A.C.-M.); 4Centro de Investigación Biomédica en Red de Cáncer (CIBERONC) (CB16/12/00400, CB16/12/00233, CB16/12/00369, CB16/12/00489 and CB16/12/00480), Instituto Carlos III, 28029 Madrid, Spain; lcsanoja-ibis@us.es (L.S.-F.); npuig@saludcastillayleon.es (N.P.); u3817@usal.es (O.G.-S.); jositojp@hotmail.com (J.P.-M.); mvmateos@usal.es (M.-V.M.); Leireburgos@gmail.com (L.B.); bpaiva@unav.es (B.P.); pilaleoz@gmail.com (P.L.); 5Institute of Biomedicine of Seville, Department of Hematology, University Hospital Virgen del Rocío of the Consejo Superior de Investigaciones Científicas (CSIC), University of Seville, 41013 Seville, Spain; 6Service of Hematology, University Hospital of Salamanca (USAL) and IBSAL, 37007 Salamanca, Spain; 7Department of Internal Medicine, University Hospital Clementino Fraga Filho, Faculty of Medicine, Federal University of Rio de Janeiro (UFRJ), Rio de Janeiro 21941-617, Brazil; robmag@hucff.ufrj.br; 8Centro de Investigación Médica Aplicada (CIMA), Instituto de Investigación Sanitaria de Navarra (IDISNA), Clínica Universidad de Navarra, 31008 Pamplona, Spain; 9Department of Immunology, Erasmus MC, (EMC) University Medical Center Rotterdam, 3015 GA Rotterdam, The Netherlands; j.temarvelde@erasmusmc.nl (J.t.M.); v.h.j.vandervelden@erasmusmc.nl (V.H.J.v.d.V.); 10Department of Hematology, Hospital General de Santa Bárbara, 42005 Soria, Spain; caguilarf@saludcastillayleon.es; 11Department of Hematology, Complejo Asistencial de Ávila, 05071 Ávila, Spain; abarez@saludcatillayleon.es; 12Department of Hematology, Complejo Asistencial de Segovia, 40002 Segovia, Spain; agarciamat@saludcastillayleon.es; 13Department of Hematology and Research Unit, Hospital Universitario de Burgos, 09006 Burgos, Spain; jlabradorg@saludcastillayleon.es; 14Department of Hematology, Hospital El Bierzo, 24404 Ponferrada, Spain; caguilera@saludcastillayleon.es; 15Cedars-Sinai Samuel Oschin Cancer Center, Los Angeles, CA 90048, USA; bdurie@myeloma.org; 16Department of Immunology, Leiden University Medical Center, 2333 ZA Leiden, The Netherlands; 17Americas Centro de Oncologia Integrado, Rio de Janeiro 22290-030, Brazil

**Keywords:** multiple myeloma, B-cell regeneration, hemodilution, measurable residual disease, immunophenotyping

## Abstract

**Simple Summary:**

B-cell regeneration during therapy has been associated with the outcome of multiple myeloma (MM) patients. However, the effects of therapy and hemodilution in bone marrow (BM) B-cell recovery have not been systematically evaluated. Here, we show that hemodilution is present in a significant fraction of MM BM samples, leading to lower total B-cell, B-cell precursor (BCP), and normal plasma cell (nPC) counts. Among MM BM samples, decreased percentages (vs. healthy donors) of BCP, transitional/naïve B-cell (TBC/NBC) and nPC populations were observed at diagnosis. BM BCP, but not TBC/NBC, increased after induction therapy. At day+100 post-autolo-gous stem cell transplantation, a greater increase in BCP with recovered TBC/NBC numbers but persistently low memory B-cell and nPC counts were found. At the end of therapy, complete response (CR) BM samples showed higher CD19^−^ nPC counts vs. non-CR specimens with no clear association between BM B-cell regeneration profiles and patient outcomes.

**Abstract:**

B-cell regeneration during therapy has been considered as a strong prognostic factor in multiple myeloma (MM). However, the effects of therapy and hemodilution in bone marrow (BM) B-cell recovery have not been systematically evaluated during follow-up. MM (n = 177) and adult (≥50y) healthy donor (HD; n = 14) BM samples were studied by next-generation flow (NGF) to simultaneously assess measurable residual disease (MRD) and residual normal B-cell populations. BM hemodilution was detected in 41 out of 177 (23%) patient samples, leading to lower total B-cell, B-cell precursor (BCP) and normal plasma cell (nPC) counts. Among MM BM, decreased percentages (vs. HD) of BCP, transitional/naïve B-cell (TBC/NBC) and nPC populations were observed at diagnosis. BM BCP increased after induction therapy, whereas TBC/NBC counts remained abnormally low. At day+100 postautologous stem cell transplantation, a greater increase in BCP with recovered TBC/NBC cell numbers but persistently low memory B-cell and nPC counts were found. At the end of therapy, complete response (CR) BM samples showed higher CD19^−^ nPC counts vs. non-CR specimens. MRD positivity was associated with higher BCP and nPC percentages. Hemodilution showed a negative impact on BM B-cell distribution. Different BM B-cell regeneration profiles are present in MM at diagnosis and after therapy with no significant association with patient outcome.

## 1. Introduction

Multiple myeloma (MM) is a hematopoietic malignancy characterized by an expansion and accumulation of (malignant) clonal plasma cells (cPC) in bone marrow (BM) and other tissues [1,2]. In most MM patients, cPC produce a clonal immunoglobulin—Ig (i.e., M-component)—which becomes detectable in blood and/or urine, in association with decreased normal residual serum Igs levels (i.e., immunoparesis) and end-organ damage (e.g., cytopenia, anemia, lytic bone lesions, renal failure) [3,4]. Introduction of novel drugs and new therapeutic options that target tumor cells with diverse effects on the patient’s immune system [5,6,7,8,9]—e.g., autologous hematopoietic stem cell transplantation (ASCT), immune modulatory drugs (e.g., lenalidomide, thalidomide and pomalidomide) together with antibody-based (daratumumab and elotuzumab) and CART-cell targeted therapies—has led to significantly higher complete response (CR) rates with prolonged progression-free survival (PFS) and overall survival (OS) [10,11]. However, the effects of such immune-modulatory therapies on the residual immune system have been less investigated.

Suppression of B-cell production and differentiation, in association with reduced levels of (normal) non-involved Igs (i.e., immunoparesis), is a hallmark of MM, which inversely correlates with disease stage and patient outcome [12,13], but this might revert back with therapy [14,15,16,17]. Conventional and novel treatment protocols for MM typically comprise sequential pulses of multiple drugs aimed at achieving and maintaining the highest quality of response [18]. However, such strategies might result in a cumulative immunosuppressive effect with an increased risk for (more pronounced) immunodeficiency and its related clinical complications [19,20,21]. Altogether, this indicates that recovery of immunoparesis after therapy for MM mostly depends on the ability of the (individual) patient hematopoietic system to regenerate the immune system, particularly the B-cell compartment [14,16,17,22,23,24,25].

Optimal B-cell and immune recovery might potentially contribute to both mitigate the clinical complications of immunoparesis and help eradicate residual tumor cells via antibody mediated mechanisms and parallel T cell responses [26,27,28]. In this regard, previous studies have shown that persistence of normal residual (n)PC in BM of MM at diagnosis [12] and after therapy is a strong favorable prognostic factor for patient outcome [17,22,29]. Similarly, patients who show long-term disease control, [17,22,25] as well as elderly (transplant-ineligible) MM patients with favorable outcomes, have been shown to display significantly higher mature B-cell counts in blood [22].

Previous studies on B-cell regeneration profiles in treated MM patients have shown the existence of highly variable profiles [12,14,15,17,22,23,24,25,29]. However, these studies mostly focused on total B-cells and nPC, frequently analyzed at a single time point prior to or after completion of therapy. To the best of our knowledge, there is no systematic description of the distribution of normal maturation-associated residual BM B-cell populations in MM during therapy that could be used as a frame of reference for better understanding the effects of therapy on B-cell regeneration and to investigate its potential association with both measurable residual disease (MRD) levels and patient outcome.

Here, we investigated the B-cell regeneration profile in 177 BM samples from 162 MM patients treated with high-dose therapy followed by ASCT at different time points during follow-up and explored its potential association with response to therapy, the BM MRD status and patient outcome.

## 2. Results

### 2.1. Distribution of Maturation-Associated B-Cell and PC Populations in Hemodiluted vs. Nonhemodiluted BM

All healthy donor (HD) BM samples showed no significant levels of hemodilution with median (range) frequencies of total B-cells of 2.6% (1%–4.6%) and nPC of 0.3% (0.08%–0.9%). Within normal BM B-cells, progressively higher median percentages of cells were observed along the B-cell maturation pathway from more immature (stage I) B-cell precursors (BCP)—0.02% (0.006%–0.2%)—and stage II BCP—0.3% (0.09%–1.5%); *p* = 0.001 vs. stage I BCP—to transitional B-cells (TBC)/naïve B-cells (NBC)—1.1% (0.3%–2%); *p* = 0.005 vs. stage II BCP—. In contrast, more mature post-germinal center (GC) memory B-cells, and both CD19^+^ and CD19^−^ nPC were less represented than TBC/NBC, and they had progressively lower median (range) numbers in BM—from the less mature B-cells to the more PC compartments: 0.4% memory B-cells (range: 0.06%–1.3%; *p* = 0.002 vs. immature/naive B-cells), 0.2% CD19^+^ nPC (0.03%–0.6%; *p* = 0.001 vs. memory B-cells), and 0.1% CD19^−^ nPC (0.02%–0.3%; *p* = 0.002 vs. CD19^+^ nPC)—(Table 1 and Figure 1A).

In contrast to normal BM, 41 of 177 (23%) BM samples from MM patients were found to be hemodiluted based on an abnormally low percentage (i.e., ≤0.002%) of CD117^hi^ mast cells (median of 0.001% in hemodiluted vs. 0.007% in non-hemodiluted BM samples; *p* ≤ 0.001), depending on the specific time point at which they had been obtained: median percentages of hemodiluted samples at diagnosis, postinduction and at day+100 after ASCT of 40% (10/25), 29% (11/38) and 18% (20/114), respectively (*p* = 0.04). As expected, hemodilution was associated with significantly lower total B-cell counts (median: 1.5% vs. 2.5% for non-hemodiluted BM; *p* = 0.006), immature stage I BCP (median: 0.04% vs. 0.1%, *p* = 0.006) and stage II BCP (median: 0.4% vs. 1.1%; *p* = 0.002), total nPC (median: 0.05% vs. 0.07%; *p* = 0.003), CD19^+^ nPC (median: 0.04% vs. 0.06%; *p* = 0.009) and CD19^−^ nPC (median: 0.003% vs. 0.007%; *p* = 0.001). In contrast, TBC/NBC were represented at similar percentages in hemodiluted vs. non-hemodiluted MM BM—median: 0.7% vs. 1.1% (*p* > 0.05) (Table 1 and Figure 1B,C)—independently of the time point at which samples had been studied (data not shown).

Overall, MRD was positive in 61% (74/121) of non-hemodiluted follow-up BM samples—median (range) MRD levels of 0.001% (≤0.0002% to 2.9%)—compared to 58% (18/31) of hemodiluted MM BM samples—median (range) MRD levels of 0.0003% (≤0.0002% to 4.8%) (*p* > 0.05)—. Similar numbers of BM cells were evaluated in both groups of samples (data not shown). Due to the effect of hemodilution on total BM B-cells and nPC, as well as on more immature BM-associated B-cell (i.e., BPC) and nPC populations, hereafter only non-hemodiluted BM samples were included in the analyses (Figure 1).

### 2.2. Distribution of Maturation-Associated B-Cell and nPC Populations in MM BM at Diagnosis and During Follow-Up

At diagnosis, MM patients showed significantly decreased median percentages of both stage I and stage II BCP in BM vs. age-matched HD: 0.0005% vs. 0.02% (*p* ≤ 0.001) and 0.01% vs. 0.3% (*p* ≤ 0.001), respectively (Figure 2). Regarding mature B-cells, significantly reduced percentages of TBC/NBC and nPC were also found in BM of MM patients studied at diagnosis vs. age-matched HD with median percentages of TBC/NBC at 0.4% vs. 1.1% (*p* = 0.01), of CD19^+^ nPC at 0.03% vs. 0.2% (*p* ≤ 0.001) and of CD19^−^ nPC at 0.006% vs. 0.1% (*p* ≤ 0.001), respectively (Figure 2).

In contrast, non-hemodiluted BM samples from treated MM patients (i.e., end of induction and day+100 after ASCT) showed an overall increased frequency of stage I and stage II BCP (*p* ≤ 0.001) vs. age-matched normal BM and vs. MM BM obtained at diagnosis (0.1% vs. 0.02% and 0.0005%, and 1.1% vs. 0.3% and 0.01%, respectively), while memory B-cell (0.04% vs. 0.4%) and nPC (0.06% vs. 0.2% and 0.007% vs. 0.1% for CD19^+^ and CD19^−^ nPC, respectively) subsets were both significantly (*p* ≤ 0.001) decreased vs. normal BM (Table 1 and Figure 1).

In more detail, significant recoveries of both stage I and stage II BCP were observed in MM BM samples obtained at the end of induction therapy vs. diagnosis, with median percentages of stage I BCP of 0.02% vs. 0.0005% (*p* ≤ 0.001) and of stage II BCP of 0.2% vs. 0.01% (*p* = 0.002), respectively (Figure 2). In contrast, (all) more mature B-cell populations remained significantly reduced at the end of induction (vs. normal BM), with a tendency toward lower median numbers than those observed at diagnosis for all B-cell and nPC populations, except for nPC populations: (i) TBC/NBC, 0.2% vs. 1.1% (*p* ≤ 0.01) and 0.4% (*p* = 0.006); (ii) memory B-cells, 0.07% vs. 0.4% (*p* ≤ 0.001) and 0.3% (*p* ≤ 0.001); (iii) CD19^+^ nPC levels, 0.05% vs. 0.2% (*p* = 0.004) and 0.03% (*p* = 0.06); and (iv) CD19^−^ nPC, 0.009% vs. 0.1% (*p* ≤ 0.001) and 0.006% (*p* = 0.85), respectively (Figure 2).

In turn, at day+100 post-ASCT, MM patients displayed a pattern consistent with BM regeneration and increased production of B-cells (Figure 2). Thus, significant (*p* ≤ 0.001) increased numbers of both stage I and stage II BCP (vs. both HD and MM BM studied at diagnosis) were observed: median of 0.2% (vs. 0.02% and 0.0005%) and of 1.3% (vs. 0.3% and 0.01%) for stage I and stage II BCP, respectively (Figure 2). In parallel, the percentage of TBC/NBC returned to levels similar to those observed in normal BM—a median of 1.5% vs. 1.1% (*p* = 0.2)—. By contrast, the number of memory B-cells remained significantly decreased vs. both normal BM and MM diagnostic samples—median of 0.03% vs. 0.4% and 0.3%, respectively (*p* ≤ 0.001)—(Figure 2). Similarly, to memory B-cell counts, the number of both CD19^+^ and CD19^−^ nPC at day+100 post-ASCT also remained below normal levels: a median of 0.07% vs. 0.2% for CD19^+^ nPC and of 0.007% vs. 0.1% for CD19^−^ nPC (*p* ≤ 0.001), respectively (Figure 2 and Appendix A).

### 2.3. Impact of Response to Therapy and Type of Induction Therapy on the Distribution of Normal Residual BM B-Cell and nPC Populations in MM

Similar distributions of the distinct subsets of BCP, mature B-cells (TBC/NBC and memory B-cells) and total nPC were observed in BM of MM patients grouped according to response to therapy (i.e., non-CR vs. sCR/CR) for the two different time points evaluated during follow-up (Table 2). In contrast, the median percentage of CD19^−^ nPC was higher in BM of patients who were in CR/sCR vs. non-CR cases, both at the end of induction—0.01% vs. 0.003% (*p* = 0.036)—and at day+100 post-ASCT—0.009% vs. 0.006% (*p* = 0.049)—. Among CR/sCR cases, the presence vs. absence of MRD did not affect the distribution of the distinct maturation-associated B-cell and nPC populations in BM studied at the end of induction (Table 2). In contrast, at day+100 post-ASCT, MRD positivity among CR/sCR patients was associated with higher median percentages (vs. MRD negativity CR/sCR patients) of: (i) total BCP (1.9% vs. 1.3%; *p* = 0.026), particularly due to increased stage II BCP counts (median of 1.7% vs. 1.1%; *p* = 0.023); (ii) post-GC B-cells (0.2% vs. 0.1%; *p* = 0.025); (iii) total nPC (0.1% vs. 0.07%; *p* = 0.048); and (iv) CD19^+^ nPC (0.09% vs. 0.06%; *p* = 0.018), but not CD19^−^ nPC (0.009% vs. 0.01%; *p* > 0.05).

From the prognostic point of view, MM patients who were MRD-positive at day+100 post-ASCT displayed significantly (*p* = 0.013) shorter PFS vs. MRD-negative cases—median PFS of 28 months vs. not reached (NR) (Figure 3A). In contrast, the distribution of the different subsets of BM BCP, mature (pre-GC and pos-GC) B-cells and nPC at day+100 post-ASCT, did not show a significant impact on PFS of MM patients, regardless of their BM MRD status (Figure 3B–D).

Regarding the type of induction therapy used prior to ASCT, MM patients that received a combination of proteasome inhibitors (PIs) plus immunomodulatory drugs (IMIDs) plus steroids presented significantly lower median percentages of (i) total B-cells (2.5% vs. 4.2%, *p* = 0.009), (ii) total BCP (1.2% vs. 1.9%, *p* = 0.013), particularly stage II BCP (1% vs. 1.8%, *p* = 0.008), and (iii) total nPC (0.05% vs. 0.1%, *p* = 0.001), especially CD19+ nPC (0.04% vs. 0.1%, *p* = 0.001) at day+100 post-ASCT, compared to those patients who received PIs plus steroids or IMIDs plus steroids and cyclophosphamide. Of note, MRD and PFS were similar between these two patient groups (Appendix A).

### 2.4. Distribution of Maturation-Associated B-Cell and PC Populations in Follow-Up BM from MM Patients According to the Cytogenetic Risk at Diagnosis

Cytogenetic risk was present in 103 patients, from whom 73 had standard risk cytogenetics and 30 showed high-risk cytogenetics. Overall, similar distributions of the distinct subsets of BCP, mature B-cells (TBC/NBC and memory B-cells) and total normal PC were observed in BM samples from MM patients according to cytogenetic risk for the two different time points evaluated during follow-up post-induction and at day+100 post-ASCT (Appendix A).

## 3. Discussion

B-cell recovery during therapy has been previously reported as a strong prognostic factor in MM [14,15,16,23,24,29,30,31] and a unique feature of MM patients who attain long-term disease control [22,25]. Despite this, the kinetics of B-cell depletion and regeneration in BM during the course of therapy have not been investigated in detail in MM. Here, we used a high-sensitive NGF approach [32] for simultaneous assessment of MRD and normal residual B-cell and PC regeneration in BM of MM patients studied at diagnosis and at different time points after starting therapy compared to age-matched healthy donors. For this purpose, the impact of hemodilution on the distribution of the distinct maturation-associated B-cell populations in BM was first evaluated used the percent BM mast cell cut-off of ≤0.002%, as previously proposed [32].

Overall, hemodilution showed a significant impact on the distribution of normal residual BCP, B-cells and nPC, as well as their subsets. Thus, significantly lower total B-cell and nPC numbers were detected in hemodiluted vs. non-hemodiluted BM samples, at the expense of decreased percentages of both stages I and II BCP and both CD19^+^ and CD19^−^ nPC, but with similar numbers of TBC/NBC. Altogether, these results indicate that BM hemodilution is associated with significantly decreased numbers of BM-derived B-cells (e.g., BCP) and nPC due to the very low numbers of both cell populations in (steady-state) adult blood [33,34], independently of the time point at which the sample was obtained during the course of disease therapy. To the best of our knowledge, this is the first study in which the impact of hemodilution on the distribution of BM B-cell and PC subsets was investigated in MM and shows that accurate analysis of the distribution of normal/residual B-cells requires the assessment of sample quality.

Several studies have previously reported on hemodilution of BM samples after MM treatment, with potential implications for MRD detection [32,35,36,37]. More recently, decreased mast cell counts in BM below the 0.002% threshold have been reported as a better marker than nucleated red cells, myeloid precursors, and B-cell precursors for identification of hemodilution of MM BM samples after therapy [32]. This is probably due to the fact that MM on one side and treatment of MM on the other side might significantly affect erythropoiesis and/or myelopoiesis [1,2]. Due to this, EuroFlow selected the mast cell-associated threshold to implement in the EuroFlow version of Infinicyt software (version 2.0, Cytognos) to indicate potential hemodilution of BM samples in MM [32]. However, normal mast cell counts in BM vary substantially among different individuals, and limited data exist on the frequency of mast cells in core BM biopsies from treated MM patients [38,39,40]. Altogether, this points out the need for new cellular markers and approaches for more accurate estimation of BM hemodilution. In this regard, some authors have used paired PB and BM samples and defined specific formulas to estimate BM hemodilution based on both absolute and/or relative cell counts as well as hemoglobin levels [34,41,42]. However, such approaches are not easy to implement in routine laboratory diagnostics, pointing out the need for more simple and user-friendly approaches to estimate BM hemodilution in individual patients.

Based on the impact of hemodilution on the B-cell and nPC distribution in BM, all subsequent analyses performed in this study were restricted to non-hemodiluted BM samples evaluable after therapy as defined by a mast cell threshold of >0.002% [32]. Overall, our results showed a distribution of the distinct B-cell subsets in normal adult (≥50y) BM, consistent with a progressive expansion of BCP along their different maturation stages, until TBC/NBC migrate to blood. In contrast, progressively lower numbers of post-GC B-cell and nPC subsets were observed in normal BM from the less mature to the more differentiated cells (i.e., memory B-cells to CD19^+^ nPC and CD19^−^ nPC). These results are fully consistent with those previously reported in the literature [32,33,43].

Compared to age-matched normal BM, MM patients presented at diagnosis with significantly decreased numbers of BCP (both stage I and stage II), TBC/NBC and (both CD19^+^ and CD19^−^) nPC in their BM. Previous studies, indicate that decreased BCP and nPC numbers in BM of MM patients at diagnosis might result from their progressive replacement by cPCs, as normal BPC, nPC and cPC share similar adhesion molecule phenotypic profiles with potentially overlapping BM (stromal cell) niches [13,44,45], whose numbers are limited [44] and functionally impaired in the elderly [46,47]. At the same time, interaction of cPC with BM stromal cells might actively induce apoptosis of B-cell progenitors, leading to decreased B-cell production and lower immature/naïve B-cell counts in blood [48].

Following induction therapy, the percentage of BCP notably increased in the BM of MM patients in parallel to the decrease in cPC percentages, suggesting the later may contribute to a greater availability of stromal cell BM niches [13], and thereby, the recovery of B-cell production. In contrast, mature pre-GC B-cells (e.g., TBC/NBC) remained significantly decreased at the end induction therapy and only recovered latter at day+100 post-ASCT, with highly variable numbers among different patients. These findings support full (but variable) recovery of B-cell production after ASCT. In contrast, at day+100 post ASCT, post-GC BM B-cells (e.g., memory B-cells) and nPC counts remained significantly decreased compared to age-matched HDs. These results together with those previously reported for MM patients who achieve long-term disease control [22,25] suggest that full recovery of memory B-cells and end-stage nPC in BM of MM might require longer periods of time (e.g., ≥1year) during which the levels of antigen-experienced B-cells (e.g., memory B-cells and end-stage nPC) in BM return to normal [44,49] and serum immunoglobulin (Ig) levels recover in the absence of disease progression [17,50]. Altogether, these results support the notion that full B-cell reconstitution is a late and progressive process that starts early after the onset of therapy, which would lead to a full recovery of normal B-cell counts at between 6 to 12 months, when maximum B-lymphocyte levels are detected in BM [30,49]. In line with these findings, the percentage of nPC in BM as assessed by flow cytometry has also been associated with parallel recovery of Ig levels from day+100 after ASCT onward [17]. Despite this, global B-cell regeneration profile, we observed important differences depending on the specific type of induction therapy administered. Thus, the use of PIs in combination with IMIDs as induction therapy appeared to more deeply affect B-cell regeneration in our patient cohort compared to the use of these same type of drugs separately. These results are in line with previous observations indicating that an increased risk of infection together with a decreased response to vaccination would occur among MM patients treated with combined (vs. single) PIs plus IMIDs therapy [19,51]. Despite the different B-cell regeneration profiles observed according to induction therapy, no significant differences in the patients’ MRD status and PFS rates were observed according to the type of induction therapy.

In contrast to the negative impact of BM infiltration by cPC on normal B-cell and nPC production at diagnosis, no major differences were observed between patients who reached sCR/CR at day+100 post-ASCT and non-CR cases, except for higher percentages of CD19^−^ nPC in the former group. In addition, no significant differences were observed between MM patients with standard vs. high cytogenetic risk. Conversely, a significantly more pronounced B-cell and nPC recovery was observed at day+100 post-ASCT among MRD-positive vs. MRD-negative sCR/CR patients. 

From the prognostic point of view, previous studies on transplant-ineligible elderly MM patients suggested an association between high BCP counts in BM and a poor outcome [24]. In contrast, for the same patient population [24] and both transplant-eligible cases [14,16,29,30,31], as well as in MM patients with long-term disease control [17,22,25], multiple studies point out better recoveries of BM B-cells and nPC have been associated with better outcomes. In line with these later studies, here we found higher numbers of CD19^−^ nPC in BM of patients who reached sCR/CR vs. non-CR cases, both at the end induction therapy and at day+100 post-ASCT. In contrast, among sCR/CR patients greater B-cell and nPC counts were observed in BM of MRD-positive vs. MRD-negative cases. Careful analysis for the potential reasons for such a discrepancy show that the majority of previous studies focused on the percentage of nPC within the whole BM PC compartment [29], the B-cell counts in blood [22,25,30,31] and Ig serum levels [14,17]. In contrast, here we focused on the relative distribution of the distinct B-cell populations in whole BM, where B-cell numbers might also be influenced by the recovery of other cell populations, including the erythroid and myeloid precursors; in addition, it is also important in this study that very limited numbers of paired BM samples from the same patients were analysed at diagnosis and follow-up, while in others potentially hemodiluted samples have been excluded from analysis. Therefore, further investigations in larger series of paired diagnostic and follow-up (non-hemodiluted) BM samples from MM patents are required to determine the underlying cause(s) for such apparent discrepancies. 

Altogether, our findings reinforce the notion that following therapy, depletion of cPCs from the BM (stromal cell) niches would increase their availability for nPC that have reached the BM and potentially also for normal BCP, leading to a recovery of both B-cell production and nPC homing in BM after therapy. However, despite previous studies suggesting that there is an association between a better recovery of more mature B-cells and an improved patient outcome, here we could not show an impact of the B-cell recovery profile on PFS of MM, even when the BM MRD status at day+100 post-ASCT did. Importantly, our data also indicate that the BM B-cell regeneration profiles in MM might also be affected by the type of induction therapy and the time point of BM collection, as well as hemodilution.

## 4. Materials and Methods

### 4.1. Patients, Samples, and Controls

A total of 177 BM samples from 162 MM patients—57% males and 43% females with a median age (range) of 62 years (y; 36–87y)—were obtained and studied at different time points during the course of disease, including: diagnosis (n = 25), end of induction therapy (n = 38) and day+100 post-ASCT (n = 114). Only 15/162 MM patients were studied sequentially at diagnosis and/or during follow-up. After therapy started, patients were categorized at every time point of evaluation by the 2016 International Myeloma Working Group (IMWG) response criteria [52] into: stringent (s)CR/CR (n = 93) and non-CR (n = 59), including cases with very good partial responses (VGPRs; n = 44), partial responses (PRs; n = 6), stable disease (SD; n = 4) and progressive disease (PD; n = 5). In addition, 14 BM samples from an identical number of healthy donors—HDs; median age (range) of 58y (50–78y)—were studied in parallel. BM samples were obtained and processed within 24h after collection at 4 different centers (USAL, UFRJ, CIMA and EMC). All BM samples from MM and HD patients were collected after each individual had given his/her informed consent to participate in the study and/or in compliance to regulations of local ethics and research committees, according to the Declaration of Helsinki; the study was approved by the local ethics committees of the four participating centers. A more detailed description of the patients’ clinical and laboratory features at diagnosis, as well as the treatment time points and therapeutic regimens used, is provided in Appendix A.

### 4.2. Treatment Regimens


At the end of induction therapy, patients had received between 4 and 6 cycles of therapy prior to ASCT. Post-ASCT (day+100), BM samples were obtained 104 ± 15 days after transplantation. Induction regimens were based on IMIDs or PIs, while high-dose melphalan was used in the conditioning regimen for ASCT with the following patient distribution per time point: (i) end of induction therapy (n = 38), included patients treated with PIs + IMIDs + steroids (n = 28), PIs + chemotherapy + steroids (n = 2), PIs + IMIDs + chemotherapy+ steroids (n = 2), PIs + IMIDs + steroids + anti-CD38 (n = 1), PIs + steroids (n = 3), chemotherapy + IMIDs + steroids (n = 1), and chemotherapy alone (n = 1); and (ii) day+100 post-ASCT (n = 114) cases consisted of MM patients previously treated with PIs + IMIDs + steroids (n = 53), PIs + cyclophosphamide + steroids (n = 23), PIs + steroids (n = 3), PIs + IMIDs + steroids + anti-CD38 (n = 2), PIs + chemotherapy + IMIDs + steroids (n = 5), cyclophosphamide + IMIDs + steroids (n = 22), IMIDs + steroids (n = 2) and chemotherapy alone (n = 1), followed in all cases by ASCT. For to the purpose to compare B-cell distribution according to the type of induction therapy, MM patients were grouped in two categories: (i) PIs + IMIDs + steroids or (ii)
PIs + steroids or IMIDs + steroids and cyclophosphamide.

### 4.3. Immunophenotypic Studies

All BM samples were stained following the EuroFlow next-generation flow (NGF) MM MRD antibody panel and standard operating procedures (SOPs), as previously described [32,53]. Briefly, bulk lysed BM samples were stained with the two 8-color EuroFlow tubes—(i) CD138-BV421, CD27-BV510, CD38-FITC, CD56-PE, CD45-PerCPCy5.5, CD19-PECy7, CD117-APC, CD81-APCC750 and (ii) CD138-BV421, CD27-BV510, CD38-FITC, CD56-PE, CD45-PerCPCy5.5, CD19-PECy7, cytoplasmic (cy)Ig—Kappa-APC and cyIg-Lambda-APCC750. Stained cells were measured on FACSCanto II flow cytometers—Becton Dickinson Biosciences (BD), San Jose, CA—using the FACS DiVA software (BD) and the EuroFlow SOPs for instrument set up and calibration [53]. A median of 5 × 10^6^ (range: 10 × 10^5^–12 × 10^6^) and 10^7^ (range: 2 × 10^6^–15 × 10^6^) cells were measured in case of diagnostic and follow-up samples, respectively. For data analysis, events measured in the two tubes stained per sample were merged into a single data file and analyzed using the automatic gating and report tool (AGI) of the Infinicyt software (version 2.0, Cytognos SL, Salamanca, Spain). Identification of the distinct maturation-associated B-cell compartments was based on the following phenotypic criteria: (i) B-cell precursors (BCP) were defined as CD19^+^ CD38^hi^ CD45^lo^ CD81^hi^ cyIg^−^ cells; (ii) TBC/NBC as CD19^+^ CD45^+^ CD38^-/+lo^ CD27^−^ CD81^+^ and cyIg^+^ cells; (iii) memory B-cells were CD19^+^ CD45^+^ CD38^−/lo^ CD27^+^ CD81^+^ cyIg^+^, and (iv) nPC were identified as CD38^hi^ CD138^lo/het^ CD45^−/+^ CD81^het^ CD27^+^ CD56^−/+^ CD117^−^ cyIg^+^ cells. In addition, BCP were further subdivided into stage I (CD27^+^) and stage II (CD27^−^) BCP as previously reported [54,55] and nPC were split according to CD19 expression into CD19^+^ plasmablasts/PC and CD19^−^ mature PC (Figure 4). For each cell population, relative distribution (i.e., percentage from all BM nucleated cells after excluding cell debris/doublets and cPC) was recorded. MRD negativity was defined as absence of cPC in BM by NGF at a limit of detection ≤2 × 10^−6^ cPC, while MRD positivity indicates presence of cPC in BM by NGF above this cut-off. Then, ≤0.002% CD117^hi^ mast cells were used to classify a BM sample as hemodiluted, following previously defined criteria [32].

### 4.4. Molecular Cytogenetic Studies

Interphase fluorescence in situ hybridization (i-FISH) studies were performed at diagnosis in 103/162 for detection of Ig heavy chain (IGH) gene rearrangements/translocations—t(4;14), t(14;16), t(14,20)—and for del(17/17p). iFISH studies were systematically performed on fluorescence-activated cell sorting (FACS)—purified cPC (FACS Aria, BD Biosciences). Based on the cytogenetic findings, patients were classified as having standard risk (n = 73) or high-risk cytogenetics (n = 30).

### 4.5. Statistical Methods

The nonparametric Kruskal–Wallis or Mann–Whitney U tests (for unpaired continuous variables) and the Wilcoxon or Friedman tests (for paired continuous variables), as well as the chi-square test (for categorical variables) were used to establish the statistical significance of differences observed among groups. PFS curves were plotted by the Kaplan and Meier method and the (two-sided) log-rank test was used to compare PFS curves. PFS was defined as the time from BM analysis by NGF to disease progression/relapse or death. To determine the impact of the BM B-cell regeneration profiles on PFS, patients were stratified based on median (percent) values observed at day+100 after ASCT for each normal residual maturation-associated B-cell population investigated. Two-sided *p*-values <0.05 were considered to be statistically significant. All statistical analyses were performed with the Statistical Package for Social Sciences Software (SPSS, version20; IBM Corp Inc, Chicago, IL, USA).

## 5. Conclusions

Hemodilution had a significant impact on the distribution of normal residual B-cells (BCP and nPC). These results reinforce the need for high-quality BM aspirate for both MRD and immune monitoring in MM after therapy. Different (altered) B-cell distribution profiles are present in MM BM at diagnosis and after therapy with no significant association with patient outcome.

## Figures and Tables

**Figure 1 cancers-13-01704-f001:**
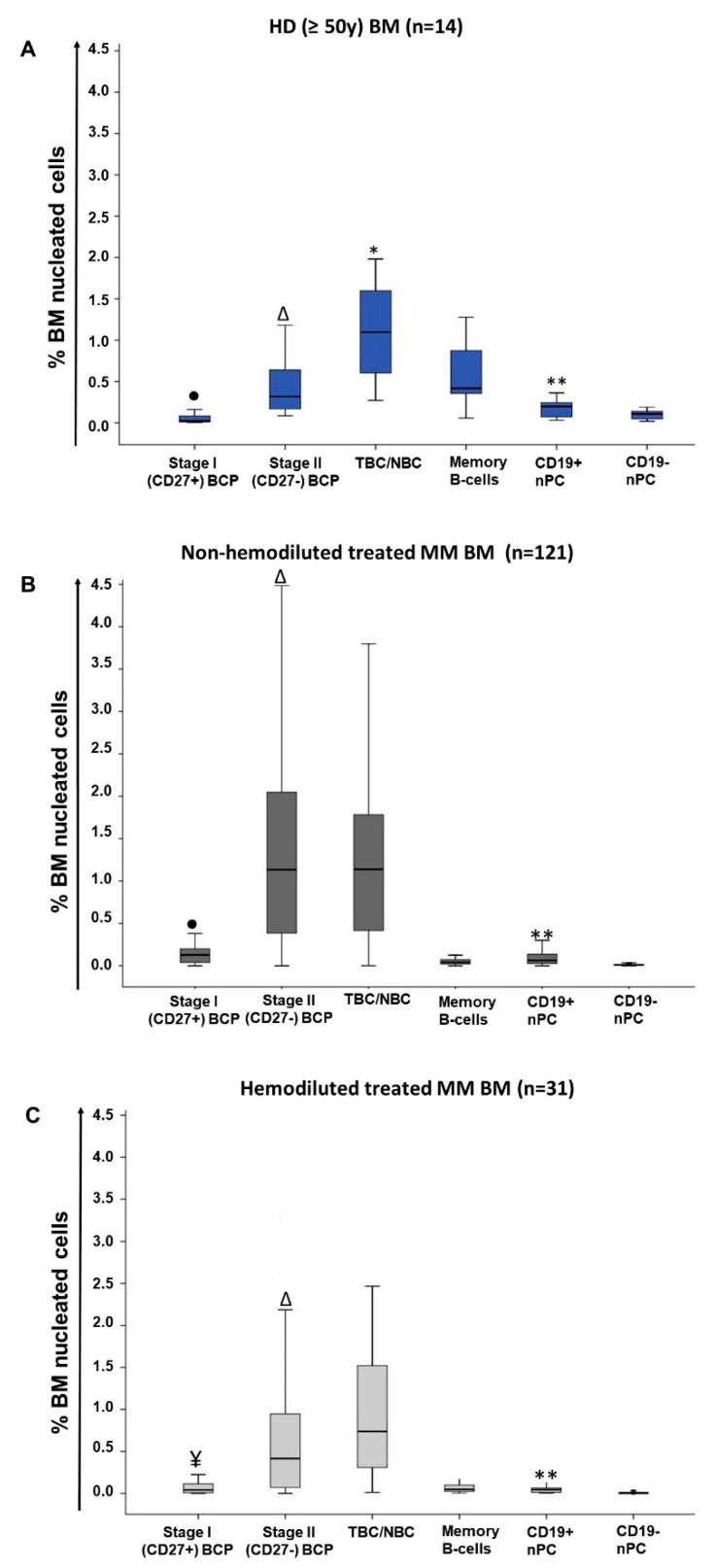
Distribution of normal/residual B-cell populations in BM of healthy adults (≥50y) (A) and both non-hemodiluted (B) and hemodiluted (C) BM samples from (treated) MM patients. Panel A: ● *p* < 0.05 vs. all B-cell and nPC populations; Δ *p* < 0.05 vs. all B-cell and nPC populations except memory B-cells; **p* < 0.05 vs. all immature and mature B-cell populations; ** *p* < 0.05 for CD19^+^ nPC vs. CD19^−^ nPC. Panel B: ● *p* < 0.05 vs. all B-cell and nPC populations; Δ *p* < 0.05 vs. all B-cell and nPC populations except transitional/naïve B-cells (TBC/NBC); ** *p* < 0.05 CD19^+^ nPC vs. CD19^−^ nPC. Panel C: ¥ *p* < 0.05 vs. all B-cell and nPC populations except memory B-cells and CD19 + nPC; Δ *p* < 0.05 vs. all B-cell and nPC populations except TBC/NBC; ** *p* < 0.05 CD19^+^ nPC vs. CD19^−^ nPC. BM, bone marrow; HD, healthy donor; MM, multiple myeloma; BCP, B-cell precursors; TBC/NBC, transitional/naïve B-cells; nPC, normal plasma cell.

**Figure 2 cancers-13-01704-f002:**
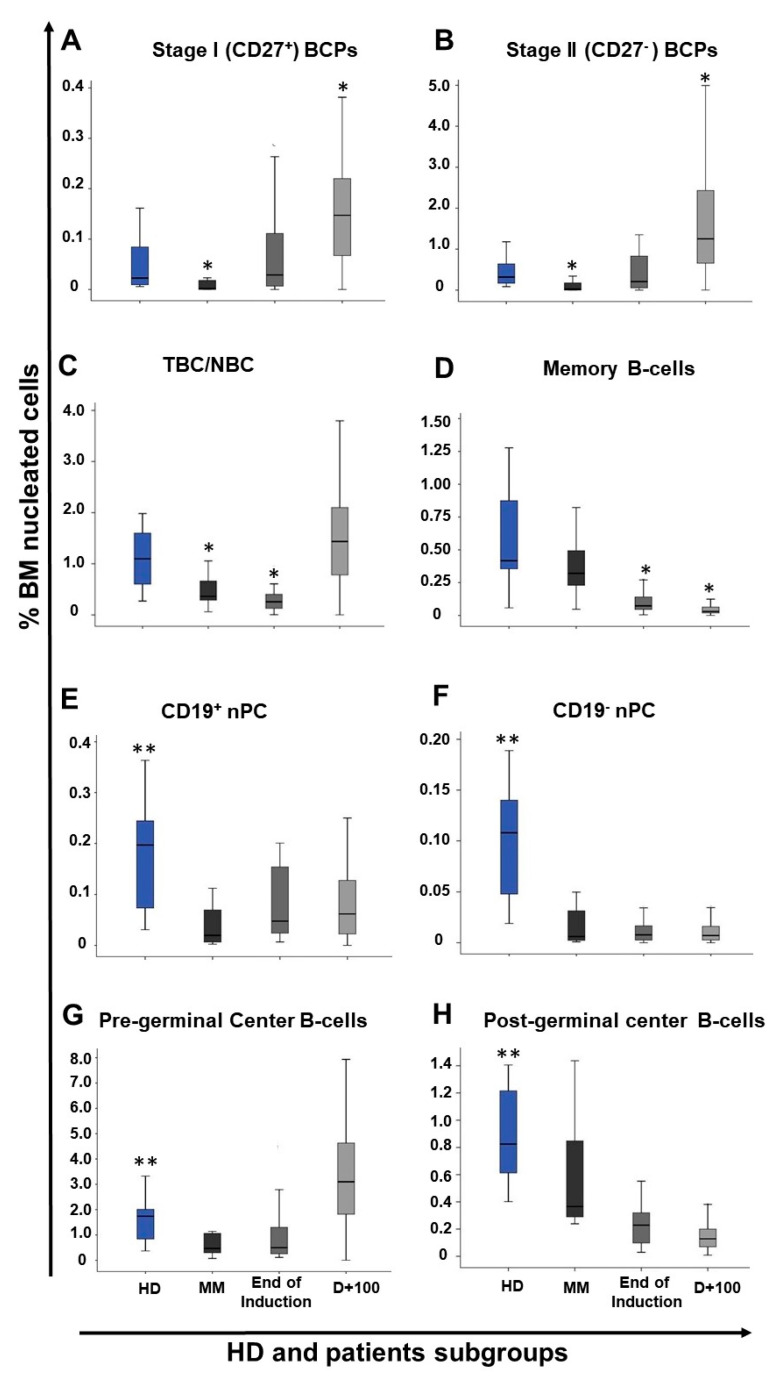
Distribution of normal/residual B-cell populations in BM samples from HDs (n = 14) vs. MM patients studied at diagnosis (n = 25) and subsequently at the end of induction therapy (n = 27) and at day+100 post autologous stem cell transplantation (ASCT) (n = 94). In panels A and B, the distributions of stage I (CD27^+^) and stage II (CD27^−^) B-cell precursors (BCP) are shown, while in panels C and D, the distributions of TBC/NBC and memory B-cells are displayed, respectively. In turn, panels E and F display the distribution in BM of CD19^+^ and CD19^−^ normal plasma cells, respectively, whereas in panels G and H the distributions of all pre-germinal center and post-germinal center B-cell populations in BM are shown, respectively. * *p* value < 0.05 vs. HD and ** *p* value <0.05 HD vs. all groups. BM, bone marrow; HD, healthy donor; BCP, B-cell precursors; TBC/NBC, transitional/naïve B-cells.

**Figure 3 cancers-13-01704-f003:**
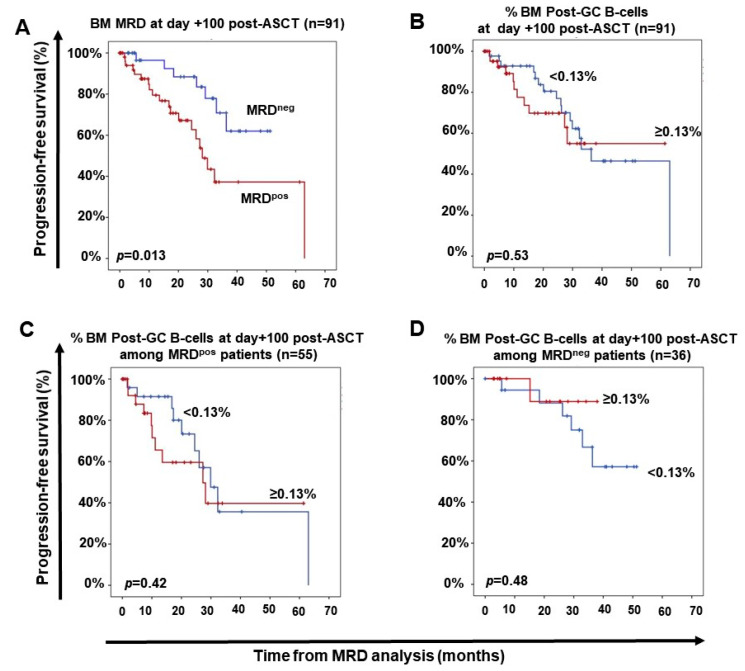
Impact of measurable residual disease (MRD) and the distribution of normal residual post-germinal center B-cells in BM of MM at day+100 after autologous stem cell transplantation (ASCT) on patient progression-free survival (PFS). In panel (**A**), the impact of BM MRD on PFS is shown. PFS of treated MM patients grouped according to the number of residual post-germinal center B-cells (memory B-cells plus nPC) in BM at day+100 post-ASCT (**B**) and in MM patients stratified according to the presence (**C**) vs. absence of MRD (**D**) at day+100 post-ASCT, is shown. BM, bone marrow; MM, multiple myeloma.

**Figure 4 cancers-13-01704-f004:**
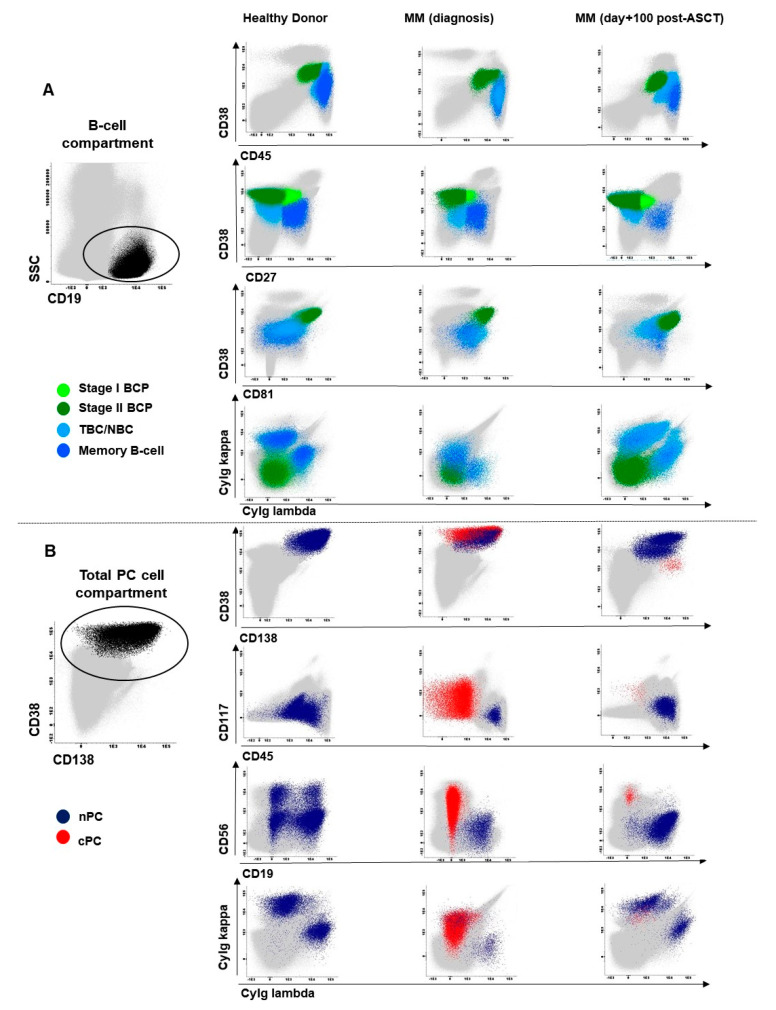
Illustration of the gating strategy used for identification of the different populations of B-cells and both normal and clonal plasma cells present in BM from a representative healthy donor, an MM patient studied at diagnosis and an infiltrated MM BM sample studied at day+100 after autologous stem cell transplantation (ASCT). In panel (**A**), total B-cells were identified as CD19^+^ and low sideward scatter (SSC) cells (left plot) and subsequently subdivided into stage I (CD27^+^) B-cell precursors (BCP; light green dots), stage II (CD27^−^) BCP (dark green dots), transitional/naïve B-cells (TBC/NBC; light blue dots), and memory B-cells (blue dots). In panel (**B**), the distributions of normal (polyclonal) plasma cells (nPC; dark blue dots) and clonal plasma cells (cPC; red dots), as defined based on their cytoplasmic (cy)Ig expression pattern, are shown. All dot plot graphical representations in the left column correspond to a representative healthy donor, while the middle and right column panels correspond to two MM patients studied at diagnosis (middle panel) and at day+100 post-ASCT (right panel). BM cells other than B-cells and PC are depicted as grey events. BM, bone marrow; MM, multiple myeloma.

**Table 1 cancers-13-01704-t001:** Distribution of maturation-associated B-cell and normal plasma cell (nPC) populations in non-hemodiluted vs. hemodiluted bone marrow (BM) from treated multiple myeloma (MM) patients (vs. healthy donor (HD) and diagnostic patient samples).

Cell Population (%)	Treated MM (n = 152)			
	HD (n = 14)	MM at Diagnosis (n = 25)	Non-hemodiluted BM (n = 121)	Hemodiluted BM (n = 31)
**Total B-cells**	**2.6** (1–4.6)	**1.2 ^a^** (0.4–3.3)	**2.5 ^b^** (0.05 -11.4)	**1.5 ^c^** (0.04–7.9)
**Pre-germinal center B-cells**	**1.7** (0.4–3.3)	**0.5 ^a^** (0.07–2.2)	**2.4 ^a,b^** (0.003–11.2)	**1.3 ^b,c^** (0.03–7.8)
BCP	**0.3** (0.09–1.7)	**0.01 ^a^** (<0.0002–0.8)	**1.3 ^a,b^** (<0.0002–10.3)	**0.5 ^b, c^** (<0.0002–6.1)
Stage I BCP	**0.02** (0.006–0.2)	**0.0005 ^a^** (<0.0002–0.1)	**0.1^a,b^** (<0.0002–0.9)	**0.04 ^b,c^** (<0.0002–0.6)
Stage II BCP	**0.3** (0.09–1.5)	**0.01 ^a^** (<0.0002–0.7)	**1.1 ^a,b^** (<0.0002–9.4)	**0.4 ^b,c^** (<0.0002–5.5)
Stage I/stage II BCP ratio	**0.1** (0.04–0.2)	**0.02 ^a^** (0–0.2)	**0.09 ^b^** (<0.0002 -1.6)	**0.1 ^b^** (<0.0002–1.2)
**Transitional/naive B-cells**	**1.1** (0.3–2)	**0.4 ^a^** (0.05–1.9)	**1.1 ^b^** (0.0008–5.7)	**0.7** (0.01–4,4)
**Post-germinal center B-cells**	**0.8** 0.4–1.4)	**0.4 ^a^** (0.1–1.6)	**0.1 ^a,b^** (0.0008–0.9)	**0.1 ^a,b^** (0.01–1)
Memory B-cells	**0.4** (0.06–1.3)	**0.3 ** (0.05–1.5)	**0.04 ^a,b^** (0.0005–0.5)	**0.05 ^a^** (0.003–1)
nPC	**0.3** (0.08–0.9)	**0.04 ^a^** (0.005–0.5)	**0.08 ^a,b^** (0.002–0.8)	**0.05 ^a,c^** (0.003–0.4)
CD19^+^ nPC	**0.2** (0.03–0.6)	**0.03 ^a^** (0.002–0.4)	**0.06 ^a,b^** (<0.0002–0.8)	**0.04 ^a,c^** (0.002–0.4)
CD19^−^ nPC	**0.1** (0.02–0.3)	**0.006 ^a^** (<0.0002–0.08)	**0.007 ^a^** (<0.0002–0.2)	**0.003 ^a,b,c^** (<0.0002–0.08)
CD19^+^/CD19^−^ nPC ratio	**2** (0.7–8.6)	**2.2 ** (0–18.6)	**7.5 ^a,b^** (0–146)	**9.4 ^a,b^** (0–45)
**Mature B-cells ¥**	**1.5** (0.4–3)	**0.8 ^a^** (0.1–2.8)	**1.2** (0.004–5.9)	**0.9 ^a^** (0.01–4.5)
**BCP/mature B-cell ratio**	**0.2** (0.08–1.6)	**0.005 ^a^** (0–2.5)	**1 ^a,b^** (0–30.2)	**0.4 ^b,c^** (0–5.9)

Abbreviations: BM, bone marrow; HD, healthy donor; BCP, B-cell precursor; nPC, normal plasma cells; MM, multiple myeloma; ¥, mature B-cells (transitional/naïve B-cells plus memory B-cells); ^a^
*p* < 0.05 vs. HD. ^b^
*p* < 0.05 vs. at diagnosis; ^c^
*p* < 0.05 for hemodiluted vs. non-hemodiluted BM (Mann–Whitney-U test).

**Table 2 cancers-13-01704-t002:** Distribution of normal/residual B-cell populations in BM of treated MM patients according to their response status studied at different time points after/during therapy.

Cell population (%)	End of Induction (n = 27)			Post ASCT (day+100) (n = 94)		
	Non-CR (n = 7)	sCR/CR MRD+ (n = 13)	sCR/CR MRD- (n = 7)	Non-CR (n = 38)	sCR/CR MRD+ (n = 27)	sCR/CR MRD- (n = 29)
**Total B-cells**	**1.3** (0.3–2.9)	**0.7** (0.3–2.1)	**0.9** (0.5–5)	**3.7** (0.06–11.4)	**4.1** (1.1–10.7)	**2.6** (0.05–9.3)
**Pre-germinal center B-cells**	**0.8 ^a^** (0.2–2.9)	**0.3 ** (0.1–1.9)	**0.7** (0.4–4.5)	**3.6** (0.05–11.2)	**3.9** (0.5–10.5)	**2.6** (0.003–9.1)
BCP	**0.5** (0.05–2.3)	**0.1** (<0.0002–1.4)	**0.3** (0.01–2)	**1.7** (0.04–10.3)	**1.9 ^b^** (0.06–9.5)	**1.3** (0.002–4.1)
Stage I BCP	**0.1** (0.005–0.3)	**0.008** (<0.0002–0.2)	**0.02** (0.001–0.7)	**0.2** (0.01–0.9)	**0.2** (0.002–0.8)	**0.1** (<0.0002–0.4)
Stage II BCP	**0.2** (0.05–2.1)	**0.1** (<0.0002–1.2)	**0.3** (0.01–1.4)	**1.5** (0.03–9.4)	**1.7 ^b^** (0.06–8.6)	**1.1** (0.002–3.8)
**Stage I/stage II BCP ratio**	**0.1** (0.07–1.6)	**0.1** (<0.0002–0.8)	**0.09** (0.03–0.5)	**0.09** (0.02–0.6)	**0.09** (0.01–0.5)	**0.08 ** (<0.0002–1.2)
**Transitional/naive B-cells**	**0.3 ^a^** (0.2–0.6)	**0.1 ** (0.004–0.4)	**0.3** (0.1–2.5)	**1.5** (0.09–5.7)	**1.3** (0.3–4.9)	**1.5** (0.0008–5.7)
**Post-germinal center B-cells**	**0.2** (0.08–0.5)	**0.2** (0.03–0.6)	**0.2** (0.08–0.6)	**0.1 ** (0.03–0.9)	**0.2 ^a,b^** (0.02–0.6)	**0.1** (0.009–0.5)
Memory B-cells	**0.08** (0.05–0.5)	**0.07** (0.005–0.5)	**0.08** (0.05–0.3)	**0.03** (0.005–0.2)	**0.03** (0.006–0.5)	**0.03** (0.0005–0.2)
nPC	**0.03** (0.01–0.2)	**0.06 ** (0.008–0.4)	**0.1** (0.04–0.4)	**0.07** (0.004–0.8)	**0.1 ^a,b^** (0.002–0.6)	**0.07** (0.008–0.3)
CD19^+^ nPC	**0.03** (0.01–0.2)	**0.05** (0.007–0.4)	**0.09** (0.03–0.4)	**0.05 ** (<0.0002–0.8)	**0.09 ^a,b^** (0.002–0.4)	**0.06** (0.0008–0.3)
CD19^−^ nPC	**0.003** (≤0.0002–0.02)	**0.01** (0.006–0.04)	**0.01** (0.003–0.07)	**0.006** (<0.0002–0.1)	**0.009** (<0.0002–0.2)	**0.01** (0.001–0.06)
CD19^+^/CD19^−^ nPC ratio	**8.4** (0–72.2)	**5.4** (2.7–19.3)	**4.8** (1.3–147)	**11.3** (0–42)	**9.5 ** (0–39)	**5** (0–21.5)
**Mature B-cells ¥**	**0.5 ^a^** (0.3–0.8)	**0.2 ** (0.009–0.7)	**0.4** (0.3–2.6)	**1.5** (0.01–5.8)	**1.4** (0.3–5)	**1.5** (0.004–5.9)
**BCP/Mature B-cell ratio**	**0.6** (0.1–5.9)	**0.6** (0–30.2)	**0.8** (0.03–3.4)	**1** (0–13.3)	**1.7** (0–8.5)	**0.9** (0.08–3.5)

Abbreviations: MM, multiple myeloma; BM, bone marrow; MRD, measurable residual disease; BCP, B-cell precursors; nPC, normal plasma cells; CR, complete response; sCR, stringent CR; ¥ Mature B-cells (transitional/naïve B-cells plus memory B-cells); ^a^
*p* < 0.05 for comparison between non-CR vs. CR/sCR plus MRD+; ^b^
*p* < 0.05 for comparison between CR/sCR plus MRD+ vs. CR/sCR plus MRD- (Mann–Whitney-U test).

## Data Availability

The data presented in this study are available on request from the corresponding author.

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
