# Peer review of "B-Cell Regeneration Profile and Minimal Residual Disease Status in Bone Marrow of Treated Multiple Myeloma Patients"

_cancers, 2021, doi:10.3390/cancers13071704_

Round 1
Reviewer 1 Report
Pontes et al. report on B-cell regeneration dynamics after therapy of multiple myeloma.
The scientific work is sound, yet the impact of the results is minor to negligible.
The present collective might have been taken to develop a flow cytometry software or script to compensatory account for hemodilution and present some more intriguing findings.
I miss some critical thoughts if the definition of hemodilution is sound. Could it be that myeloma bone marrows contain per se less mast cells? Shouldn't the number of erythrocytes be taken also into account?
I miss information of the type of induction therapy; this is important since there are several types of possible induction therapies with potentially different effects on B-cell regeneration, which should be taken into account.
I miss information on the cytogenetics of the myeloma cases. Since various cytogenetic constellation variably respond to induction therapy, this should be taken into account.
Author Response
REVIEWER #1
Comment 1.- The scientific work is sound, yet the impact of the results is minor to negligible.
Answer to comment 1.- We thank the reviewer for the critical evaluation of the paper contributions. Following the reviewer’s comment, we have added a new sentence at the end of the discussion section of the revised manuscript in which the paper contributions are summarized. Discussion section: Page 12, lines 415-418.
Comment 2.- The present -collective might have been taken to develop a flow cytometry software or script to compensatory account for hemodilution and present some more intriguing findings.
Answer to comment 2.- In fact, an alarm related to the absence or decreased numbers of mast cells in bone marrow of multiple myeloma patients has been added in the EuroFlow version of Infinicyt software, which once validated will be worldwide available for flow cytometrists. This is now specified in the discussion section of the revised manuscript. Discussion section: Page 11, lines 316-327.
Comment 3.- I miss some critical thoughts if the definition of hemodilution is sound. Could it be that myeloma bone marrows contain per se less mast cells? Shouldn't the number of erythrocytes be taken also into account?
Answer to comment 3.- In line with the comment of the reviewer, a new sentence has been added in the Discussion section of the new revised version of the manuscript, as regards the potential advantages and particularly, the limitations of the use of mast cell counts as an indicator of hemodilution of (follow-up) aspirated BM samples in multiple myeloma. Discussion section: Page 11, lines 317-333.
Comment 4.- I miss information of the type of induction therapy; this is important since there are several types of possible induction therapies with potentially different effects on B-cell regeneration, which should be taken into account.
Answer to comment 4.- The type of induction therapy used is now specified in the material and methods section of the new revised version of the manuscript, and new sentences on the impact of the type of induction therapy on the results have been also added both in the Results and Discussion sections of the revised manuscript.
Material and Methods section 4.2, page 13, lines 442 and 457; Results section 2.3, page 8, lines 242-249; Discussion section page 12, lines 372-381 and Supplementary Table S4, page 4.
Comment 5.- I miss information on the cytogenetics of the myeloma cases. Since various cytogenetic constellation variably respond to induction therapy, this should be taken into account.
Answer to comment 5.- Data on patient cytogenetics and its correlation with the B cell regeneration kinetics have been added in the new revised version of the manuscript (Results section) and appropriately discussed in a new sentence included also now in the Discussion section of the paper. In addition, the information on how cytogenetics data was obtained and patients classified based on it, has been added in a new sentence introduced in the Material and methods section of the manuscript. Material and methods, section 4.3, page 15, lines 522-528; Results section 2.4, page 8, lines 251-259; Discussion sections, page 12, lines 385-386 and Supplementary Table S3, page 3.

Reviewer 2 Report
The authors evaluated B-cell compartment and regeneration in patients with multiple myeloma at diagnoses and at follow-up post induction therapy and 100-day post autologous transplant, in comparison with healthy population. Their findings showed a better reconstitution at last time-point, although it did not reach the pattern observed in healthy population and without a correlation with disease response.
From what I got from text and table, it was not the same cohort of patients investigated at diagnosis and than followed-up after therapy, and this might be a weak point of the study
Few additional observations:
- the meaning of HD should be made explicit in the main text also
- please, specify the type of induction therapy used. If different among patients, would the class of drugs (i.e. monoclonal antibodies vs IMIDs vs proteasome inhibitors) impact on B cell reconstitutions?
- It is well known that aspirate hemodilution impaired the correct meaning of the results, I do not get why the authors stressed this so much
- would be more helpful use absolute numbers instead of cellular %?
Author Response
REVIEWER #2:
General comment. - The authors evaluated B-cell compartment and regeneration in patients with multiple myeloma at diagnoses and at follow-up post induction therapy and 100-day post autologous transplant, in comparison with healthy population. Their findings showed a better reconstitution at last time-point, although it did not reach the pattern observed in healthy population and without a correlation with disease response.
Answer to the general comment. - We thank the reviewer for the accurate and concise report about the paper contents and we have nothing to add in this regard.
Specific comments and observations:
Comment 1.- From what I got from text and table, it was not the same cohort of patients investigated at diagnosis and then followed-up after therapy, and this might be a weak point of the study.
Answer to comment 1.- Following the comment of the reviewer, it is now clearly specified in the material and methods section of the revised manuscript, how many patients were studied at diagnosis and follow-up. Following the reviewer’s comment, a sentence of caution has also been added in this regard in the text of the Discussion section of the new revised version of the manuscript. Material and Methods, page 13, lines 425-426; Discussion section: Page 12, lines 403-407.
Comment 2.- The meaning of HD should be made explicit in the main text also.
Answer to comment 2.- The meaning of the HD (healthy donor) abbreviation for healthy donor is now specified explicitly, the first time it appears in the text of the manuscript, as well as in all Figure legends where it is used.
Comment 3.- Please, specify the type of induction therapy used. If different among patients, would the class of drugs (i.e. monoclonal antibodies vs IMIDs vs proteasome inhibitors) impact on B cell reconstitutions?
Answer to comment 3.- The type of induction therapy used is now specified in the material and methods section of the new revised version of the manuscript, following the indication of the reviewer. In addition, new sentences on the impact of the type of induction therapy on the results have been added both in the Results and Discussion sections of the revised manuscript.
Material and Methods section 4.2, page 13, lines 442 and 457; Results section 2.3, page 8, lines 242-249; Discussion section page 12, lines 372-381 and Supplementary Table S4, page 4.
Comment 4.- It is well known that aspirate hemodilution impaired the correct meaning of the results, I do not get why the authors stressed this so much.
Answer to comment 4.- Following the comment of the reviewer, we have toned down (and even deleted some of) the statements about the impact of hemodilution on the results obtained throughout the manuscript. Discussion section: Page 11, lines 311-315 excluded.
Comment 5.- Would be more helpful use absolute numbers instead of cellular %?
Answer to comment 5.- We fully agree with the comment of the reviewer in that absolute counts might help. However, these were not obtained for bone marrow samples as it is not a standard procedure and thereby, can not be added retrospectively. Despite this, we have added a new sentence in the text of the Discussion section of the manuscript in which the potential contribution of absolute counts to assess hemodilution in BM aspirated samples is discussed. Discussion section: page 11, lines 325-333.

Reviewer 3 Report
The paper written by the authors are well presented to report on the hemodilution after myeloma treatment, it's rare reported. Through the panel analysis, we can see the authors clearly stated that hemodilution showed a negative impact on BM B-cell distribution, however, this profile showed no significant association with patients' outcome in clinic. Since more and more immunotherapy drugs have been or will be applied in MM, this study still leads us to focus on the B cell distribution during disease progress and treatment.
Author Response
REVIEWER #3:
General comment. - The paper written by the authors are well presented to report on the hemodilution after myeloma treatment, it's rare reported. Through the panel analysis, we can see the authors clearly stated that hemodilution showed a negative impact on BM B-cell distribution, however, this profile showed no significant association with patients' outcome in clinic. Since more and more immunotherapy drugs have been or will be applied in MM, this study still leads us to focus on the B cell distribution during disease progress and treatment.
Answer to the general comment. - We thank the reviewer for the brief summary of the paper contents and his positive overall assessment. We have nothing to add to the reviewer statements.

Round 2
Reviewer 1 Report
The authors addressed my previous issues. Thank you.